# Provable Defenses against Spatially Transformed Adversarial Inputs: Impossibility and Possibility Results

## Abstract

One intriguing property of neural networks is their inherent vulnerability to adversarial inputs, which are maliciously crafted samples to trigger target networks to misbehave. The state-of-the-art attacks generate adversarial inputs using either pixel perturbation or spatial transformation. Thus far, several provable defenses have been proposed against pixel perturbation-based attacks; yet, little is known about whether such solutions exist for spatial transformation-based attacks. This paper bridges this striking gap by conducting the first systematic study on provable defenses against spatially transformed adversarial inputs. Our findings convey mixed messages. On the impossibility side, we show that such defenses may not exist in practice: for any given networks, it is possible to find legitimate inputs and imperceptible transformations to generate adversarial inputs that force arbitrarily large errors. On the possibility side, we show that it is still feasible to construct adversarial training methods to significantly improve the resilience of networks against adversarial inputs over empirical datasets. We believe our findings provide insights for designing more effective defenses against spatially transformed adversarial inputs.

## 1 Introduction

Despite their tremendous success in computer vision and pattern recognition (LeCun et al., 2015), neural networks are inherently vulnerable to adversarial inputs – those maliciously crafted samples to trigger target networks to misbehave – which hinders their application in security-critical domains (Szegedy et al., 2014; Goodfellow et al., 2015; Sharif et al., 2016; Eykholt et al., 2017). This serious vulnerability has spurred intensive research effort, leading to an arms race between developing more powerful attacks (Szegedy et al., 2014; Moosavi-Dezfooli et al., 2015; Papernot et al., 2016a; Carlini & Wagner, 2017; Meng & Chen, 2017; Xu et al., 2018) and designing more robust defenses (Papernot et al., 2016b; Madry et al., 2018; Tramèr et al., 2018; Kannan et al., 2018). For example, defensive distillation (Papernot et al., 2016b), originally considered as an effective defense, was later shown to be ineffective against stronger attacks (Carlini & Wagner, 2017).

To end this constant arms race, several provable defense methods (Bastani et al., 2016; Raghunathan et al., 2018; Wong & Zico Kolter, 2018) have been proposed recently. Motivated by that most existing attacks generate adversarial inputs by directly modifying the pixel values of benign inputs, where the perturbation "imperceptibility" is often quantified by its $L_p$ norm, such provable defenses, by reducing the upper bound of the worst-case loss that can be caused by norm-bounded perturbation, provides the following guaranteed protection: for a given network and test input, no attack is able to force the error to exceed a certain threshold.

The effectiveness of these provable defenses hinges on modeling the upper bound of the worst-case loss under norm-bounded perturbation. While most existing attacks indeed use $L_p$ norm to quantify the perturbation imperceptibility, it has long been recognized that this is not an ideal metric (Johnson et al., 2016; Isola et al., 2016). Recently, spatial transformation has been proposed as a new approach for generating adversarial inputs (Xiao et al., 2018; Engstrom et al., 2017b): instead of directly modifying pixel values, it changes their spatial positions (e.g., via rotation and shifting), which is shown to better preserve the identity and structure of the original image (Zhou et al., 2016). It is observed that spatial transformation-based adversarial inputs are often perceptually less distinguishable from pixel perturbation-based counterparts, as shown in Figure1. However, for spatial transformation, it is insensible to measure the perturbation imperceptibility using $L_p$ norm; rather, alternative metrics such as the total variation (Rudin et al., 1992) are used to quantify

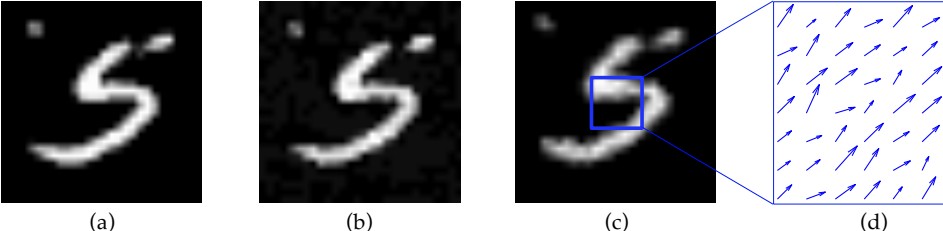

Figure 1: (a) benign input, (b) pixel perturbation-based adversarial input, (c) spatial transformation-based adversarial input, and (d) flow vector of spatial transformation.

the magnitude of spatial deformation. In other words, spatially transformed adversarial inputs are inherently not norm-bounded, which thus raises an important and intriguing question:

> *Does there exist provable defenses that provide a given network with guaranteed protection against spatially transformed adversarial inputs?*

In this paper, we present a systematic investigation to answer this question. Under the setting of a neural network with one hidden layer, we conduct both analytical and empirical studies on the existence of provable defenses. Our findings convey mixed messages. The impossibility results indicate that such defenses may not exist in practice. We show that for *any* given networks, it is possible to find benign inputs and imperceptible transformations to produce adversarial inputs that force arbitrarily large errors. Meanwhile, the possibility results imply that while the worst case can be arbitrarily bad, it is still possible to construct adversarial training methods to significantly improve the robustness of given networks over empirical datasets.

In summary, to our best knowledge, this work represents the first in-depth study on provable defenses against spatial transformation-based adversarial attacks. We provide both impossibility and possibility results regarding the existence of such defenses. We believe that our findings will highlight the fundamental difference of spatial transformation- and pixel perturbation-based attacks from the defender's perspective, and inspire designing more effective defenses against spatially transformed adversarial inputs.

## 2 BACKGROUND

### 2.1 SPATIAL TRANSFORMATION

We model a network as a function $f$. For simplicity, we focus on a binary classification task ('+' and '-') (the extension to the multi-class setting given in the appendix), in which the network predicts the class of a given input $x$ as $f(x) > 0$ as '+' and $f(x) < 0$ as '-'. In adversarial attacks, the adversary crafts an adversarial input $\tilde{x}$ by applying perturbation $r$ to a benign input $x$: $\tilde{x} = g(r; x)$, with the objective of forcing $\tilde{x}$ to be misclassified. To maximize the attack evasiveness, the adversary also desires to preserve $x$'s perceptual quality in $\tilde{x}$.

According to their perturbation types, existing adversarial attacks can be categorized as either pixel perturbation-based attacks or spatial transformation-based attacks. Figure 1 compares two versions of adversarial inputs with respect to the same benign input.

In spatial transformation attacks, instead of directly modifying $x$'s pixel values, the adversary applies a spatial transformation (e.g., shifting and rotation) over $x$ to generate $\tilde{x}$. Specifically, we use the per-pixel flow field (displacement) $r$ to synthesize $\tilde{x}$ using the pixels of $x$. Let $\tilde{x}_i$ be $\tilde{x}$'s $i$-th pixel and $(\tilde{u}_i, \tilde{v}_i)$ be its position in $\tilde{x}$. We optimize the amount of displacement with a flow vector $r_i :=$ $(\Delta u_i, \Delta v_i)$, which goes from $\tilde{x}$ to $x$. The position of $\tilde{x}_i$'s corresponding pixel in $x$ is derived as: $(u_i, v_i) = (\tilde{u}_i + \Delta u_i, \tilde{v}_i + \Delta v_i)$. As $(u_i, v_i)$ may be fractional numbers and do not necessarily lie on the integer grid, we use the differentiable bilinear interpolation to compute $\tilde{x}_i$'s pixel value:

$$\tilde{x}_i = \sum_{j \in \mathcal{N}(i)} x_j \max\left(0, 1 - |\tilde{u}_i + \Delta u_i - u_j|\right) \max\left(0, 1 - |\tilde{v}_i + \Delta v_i - v_j|\right) \quad (1)$$

where $j$ iterates over $\mathcal{N}(i)$: the set of pixels adjacent to $(u_i, v_i)$ (top-left, top-right, bottom-left, and bottom-right) in $x$. Under this setting, $\ell_{\text{diff}}$ can be instantiated with the total variation Rudin et al. (1992), which compares the spatial movement distance for any two adjacent pixels:

$$\ell_{\text{diff}}(r) = \sqrt{\sum_i \sum_{j \in \mathcal{N}(i)} (||\Delta u_i - \Delta u_j||_2^2 + ||\Delta v_i - \Delta v_j||_2^2)} \tag{2}$$

where $i$ iterates over all the pixels.

We now derive its matrix form. Let $r \in \mathbb{R}^{d \times 2}$: $r = [r^{(u)}, r^{(v)}]$ be the flow field, where $r^{(u)}$ and $r^{(v)}$ respectively represent the transformation along the horizontal and vertical directions. We assume $r^{(u)} = r^{(v)}$ and will justify this assumption in § 3. With a little abuse of notations, let $r = r^{(u)} = r^{(v)}$. Let $M^{(t)}$ be the top neighbor matrix, with the $i$-th row being the top neighbor $j$ of the $i$-th pixel (i.e., the one-hot encoding of $j$). Similarly, we define $M^{(b)}$, $M^{(l)}$, and $M^{(r)}$ respectively as the bottom, left, and right neighbor matrices. We have the proposition (proof in appendix):

**Proposition 1.** *The magnitude of spatial transformation can be specified as:*

$$\ell_{\text{diff}}(r) = \sqrt{2r^\top M^\top M r}$$

*where $M^\top M = \sum_{i \in \{t,b,l,r\}} \left(I - M^{(i)}\right)^\top \left(I - M^{(i)}\right)$.*

Therefore, the constraint on the spatial transformation magnitude can be given as:

$$\ell_{\text{diff}}(r) = \|Mr\| \leq \epsilon \tag{3}$$

## 2.2 ADVERSARIAL LOSS

Let $g(r; x)$ be the adversarial attack that applies the transformation $r$ to $x$ to generate the adversarial input $\tilde{x}$, i.e., $\tilde{x} = g(r; x)$ and $x = g(0; x)$. Therefore, for a given benign input $x$, we define the margin between $f(\tilde{x})$ and $f(x)$ as the *adversarial loss*. Formally,

$$\ell_{\text{adv}}(x, r) \overset{\text{def}}{=} \frac{f(\tilde{x}) - f(x)}{2} = \frac{f \circ g(r; x) - f(x)}{2}$$

Note that $\ell_{\text{adv}}(x, r)$ is a function of $r$.

In the case of bilinear interpolation as defined in Eqn (1), $g$ is differentiable with respect to $r$. Let $g_j$ be the $j$-th element of $g$ and $r_i$ be $r$'s $i$-th element. Note that in Eqn (1), $\tilde{x}_i$ only depends on $r_i$. Thus, $\frac{\partial g_j}{\partial r_i} = 0$ for $i \neq j$. Meanwhile,

$$\frac{\partial g_i}{\partial r_i} = \sum_j x_j \max\left(0, 1 - |\tilde{v}_i + r_i - v_j|\right) \begin{cases} 0 & |\tilde{u}_i + r_i - u_j| \geq 1 \\ 1 & \tilde{u}_i + r_i \geq u_j \\ -1 & \tilde{u}_i + r_i < u_j \end{cases}$$

where sub-gradients are used (Jaderberg et al., 2015) due to the discontinuities of interpolation.

The Jacobian $\mathrm{D}g$ of $g$ is a diagonal matrix with its $i$-th diagonal element given by $\frac{\partial g_i}{\partial r_i}$. Further, the gradient of $f$ with respect to $g$ is given as: $\nabla f = \left[\frac{\partial f}{\partial g_1}, \frac{\partial f}{\partial g_2}, \ldots, \frac{\partial f}{\partial g_d}\right]^\top$.

Putting everything together using the chain rule, we have the gradient of $f \circ g$ with respect to $r$ as:

$$\nabla(f \circ g)(r; x) = (\mathrm{D}g)^\top \nabla f$$

We now compute $\ell_{\text{adv}}(r; x)$ using the integration along the line from $x$ to $\tilde{x} = f \circ g(r; x)$:

$$\ell_{\text{adv}}(x, r) = \int_0^1 \left(\nabla(f \circ g)(zr; x)\right)^\top r \mathrm{d}z \tag{4}$$

## 3 IMPOSSIBILITY RESULTS

Next we present the negative results for provable defenses against spatially transformed adversarial inputs. Intuitively, we show that given a classifier $f$ and a threshold $\delta > 0$, if the perturbation magnitude is allowed to be *reasonably* large, the adversary is able to find an input $x_*$ and its corresponding

spatial transformation $r_*$ such that the adversarial input causes $f$ to misclassify the adversary input $g(r_*; x_*)$ with a large margin $\ell_{\mathrm{adv}}(x_*, r_*) \geq \delta$.

Specifically, we consider the maximum influence of spatial transformation on the adversarial loss via the following supremum:

$$\sup_{\|x\|_\infty \leq 1, \|Mr\| \leq \epsilon} \ell_{\mathrm{adv}}(x, r)$$

We then find a particular input-transformation pair $(x_*, r_*)$ such that $f \circ g(r_*; x_*) - f(x_*)$ represents a lower bound of this supremum. Apparently, if $f \circ g(r_*; x_*) - f(x_*) \geq 2\delta$, the supremum of $\ell_{\mathrm{adv}}(x, r)$ must exceed $\delta$. Next we present the concrete attack for a network with one hidden layer.

## 3.1 LOWER BOUND ATTACK

For a network with one hidden layer, we have $f(x) = v^\top \sigma(Wx)$ where $W$ and $v$ respectively represent the weights of the first and second layers of the network, and $\sigma$ is the activation function (e.g., ReLU). In this case, we have $\nabla f = W^\top \mathrm{diag}(v)\nabla\sigma$, where $(\nabla\sigma)_i = \frac{\partial \sigma(y_i)}{\partial y_i}$ with $y_i = (Wg(r; x))_i$. Substituting it into Eqn (4), we have:

$$\ell_{\mathrm{adv}}(x, r) = \int_0^1 (\nabla\sigma)^\top \mathrm{diag}(v) W \mathrm{D}g(zr)r\mathrm{d}z \tag{5}$$

We consider a linear approximation of $\ell_{\mathrm{adv}}(x, r)$:

$$\tilde{\ell}_{\mathrm{adv}}(x, r) = (\nabla\sigma(Wx))^\top \mathrm{diag}(v) W \mathrm{diag}(x)r$$

The following theorem finds a lower bound of the supremum of $\tilde{\ell}_{\mathrm{adv}}(x, r)$ (proof in appendix).

**Theorem 1.** *For a given network $f$, assuming $W$ has rank $d$ and $W_d$ is a rank-$d$ submatrix of $W$,*

$$\sup_{\|x\|_\infty \leq 1, \|Mr\| \leq \epsilon} \tilde{\ell}_{\mathrm{adv}}(x, r) \geq \epsilon \|M^{-1} \mathrm{diag}(x_*) W^\top \mathrm{diag}(v) \nabla\sigma(Wx_*)\|$$

*where $x_* = \pi_\infty(W_d^{-1}\mathbf{1})$ ($\mathbf{1}$ is an all-one vector). The lower bound is attained when*

$$r_* = \epsilon M^{-1} \pi_2(M^{-1} \mathrm{diag}(x_*) W \mathrm{diag}(v) \nabla\sigma(Wx_*)). \tag{6}$$

Theorem 1 gives the initial setting of $(x_*, r_*)$. We further increase the adversarial loss by improving both $x_*$ and $r_*$. Let $r^{(i)} = \frac{(2i-1)}{2n}r$. We compute the midpoint approximation of Eqn (5):

$$\ell_{\mathrm{adv}}(x, r) \approx \frac{1}{n} \underbrace{\sum_{i=1}^n \nabla\sigma\left(Wg\left(r^{(i)}; x\right)\right) \mathrm{diag}(v) W \mathrm{D}g\left(r^{(i)}; x\right) r}_{\text{Summation}} \tag{7}$$

where we divide the interval $[0, 1]$ into $n$ intervals.

**Updating $x_*$** To update $x_*$, we consider $x$ as a symbolic vector. Yet, it is difficult to optimize Eqn (7) with respect to $x$ directly as $\nabla\sigma$ also depends on $x$. Instead, we update $\nabla\sigma$ and $x$ alternatively.

Let $x_k$ be the current setting of $x_*$. We update $x_*$ as follows. Let $\alpha_i$ be the coefficient of $x$'s $i$-th element in Eqn (7) with $\nabla\sigma$ computed based on $x_k$. We set $x_{k+1} = [\mathrm{sign}(\alpha_1), \mathrm{sign}(\alpha_2), \ldots, \mathrm{sign}(\alpha_d)]^\top$, which maximizes Eqn (7) under fixed $r$, fixed $\nabla\sigma$ and the constraint of $\|x\|_\infty \leq 1$. We repeat this procedure until convergence.

**Updating $r_*$** Meanwhile, for fixed $x_*$, we may also increase the adversarial loss by refining $r_*$ using the integration approximation. Let $r_k$ be the current setting of $r_*$. We update $r_*$ using the rule of:

$$r_{k+1} = \epsilon M^{-1} \pi_2 \left( M^{-1} \sum_{i=1}^n \mathrm{D}g\left(r_k^{(i)}; x\right) W \mathrm{diag}(v) \nabla\sigma\left(Wg\left(r_k^{(i)}; x\right)\right) \right) \tag{8}$$

Intuitively, we find $r_{k+1}$ that aligns with the summation in Eqn (7) under $\|Mr_{k+1}\| \leq \epsilon$.

Algorithm 1 sketches the attack against a given network. After initialization (line 1-3), it updates $x_*$ and $r_*$ alternatively until convergence (line 4-8).

---

**Algorithm 1:** Lower Bound Attack.

**Input**: $v$, $W$: $f$'s weights; $\epsilon$: threshold of perturbation magnitude; $M$: neighboring matrix; $\alpha$: update step; $n$: parameter of midpoint approximation

**Output**: $x_*$: genuine input; $r_*$: spatial transformation

```
// initialization
```
1  $W_d \leftarrow$ rank-$d$ submartrix of $W$;
2  $x_* \leftarrow \pi_\infty(W_d^{-1}\mathbf{1})$;
3  initialize $r_*$ according to Eqn (6);
4  **while** *not converged* **do**
```
        // refinement of x_*
```
5      **while** *not converged* **do**
6          compute Eqn (7) with symbolic $x$;
```
            // α_i: coefficient of x's i-th element in Eqn (7)
```
7          $x_* \leftarrow [\text{sign}(\alpha_1), \text{sign}(\alpha_2), \dots, \text{sign}(\alpha_d)]^\top$;
```
        // refinement of r_*
```
8      update $r_*$ according to Eqn (8);
9  return $(x_*, r_*)$;

---

### 3.2 EMPIRICAL EVALUATION

Next we empirically validate the above analytical result. We mainly use MNIST (LeCun et al., 1998) as the benchmark dataset and a network with one hidden layer (with 512 hidden neurons) as the target model $f$. We consider three variants of $f$: (i) $f_{\text{nt}} - f$ is normally trained, (ii) $f_{\text{rd}} - f$ is randomly initialized, and (iii) $f_{\text{at}} - f$ is adversarially trained (Madry et al., 2018).

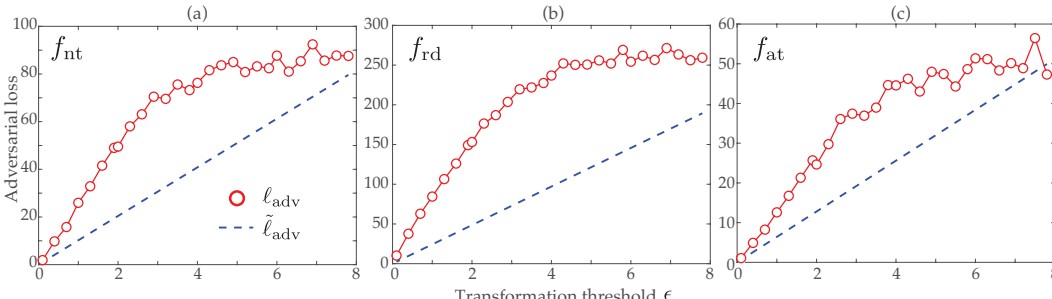

Figure 2: Lower bounds of adversarial loss by Algorithm 1 (w.r.t. $\ell_{\text{adv}}$) and Theorem 1 (w.r.t. $\tilde{\ell}_{\text{adv}}$).

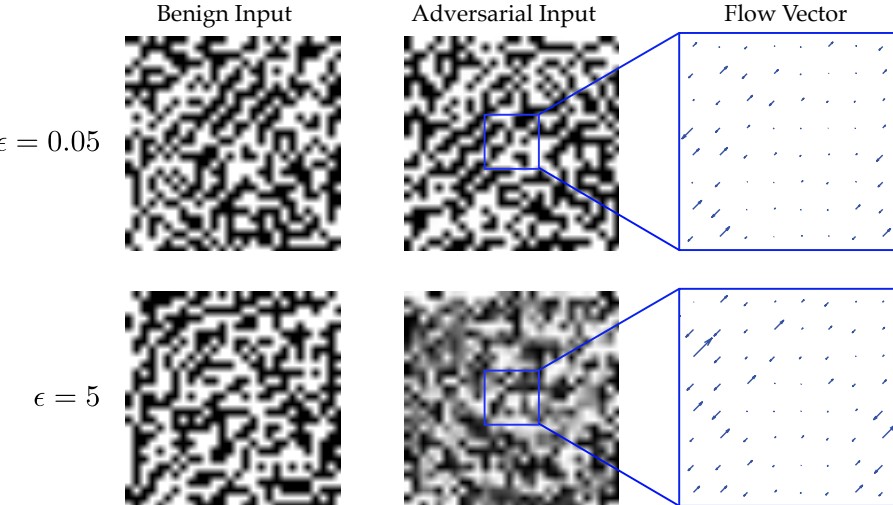

Figure 3: Samples of benign and adversarial inputs found by Algorithm 1.

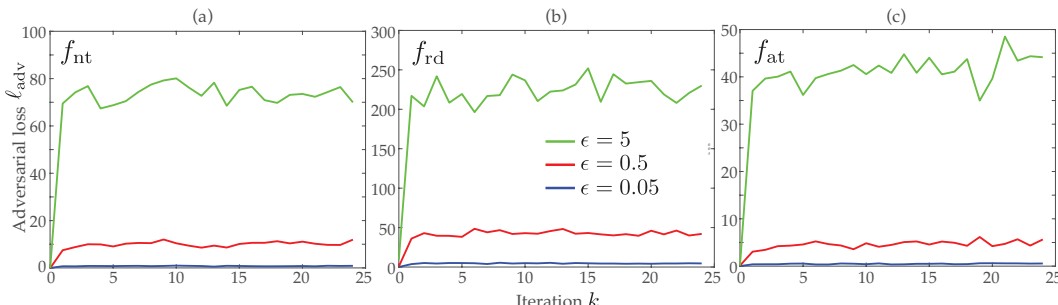

Figure 4: Convergence of estimation of adversarial loss by Algorithm 1.

We first measure the quality of lower bounds given by Algorithm 1. With respect to each of the networks, under different setting of $\epsilon$, we measure the estimation by Algorithm 1 (with respect to $\ell_{\mathrm{adv}}$) and that by Theorem 1 (with respect to $\tilde{\ell}_{\mathrm{adv}}$). Figure 2 shows the results. It is clear that as $\epsilon$ varies from 0 to 7.84, the adversarial loss increases in an exponential manner, indicating that with reasonably transformation magnitude, it is possible to find $(x_*, r_*)$ that forces arbitrarily large error. Also note that Algorithm 1 provides much higher-quality estimation of adversarial loss compared with the initial estimation by Theorem 1. Figure 3 shows a set of samples of benign and adversarial inputs (and their associated flow fields).

Figure4 shows that Algorithm 1 typically converges fast under varied setting of network and $\epsilon$, which validates our analysis of the performance of Algorithm 1.

## 4 POSSIBILITY RESULTS

Despite the impossibility results for provable defenses against spatial transformation-based attacks, in this section, we show that it is still possible to construct adversarial training methods that significantly improve DNN resilience on empirical datasets.

### 4.1 ADVERSARIAL TRAINING

To this end, we establish an upper bound on $\ell_{\mathrm{adv}}(x, r)$ and reduce this upper bound in training $f$ to improve its robustness. Specifically, we consider the integration form of $\ell_{\mathrm{adv}}(x, r)$ in Eqn (4).

$$\ell_{\mathrm{adv}}(x, r) = \int_0^1 (\nabla(f \circ g)(zr))^\top r \mathrm{d}z$$

$$\leq \max_{z \in [0,1], \|Mr\| \leq \epsilon} (\nabla(f \circ g)(zr; x))^\top r$$

In the following, we derive its upper bound form for a network with one hidden layer.

We have the following derivation regarding $\ell_{\mathrm{adv}}(x, r)$:

$$\ell_{\mathrm{adv}}(x, r) \leq \max_{\nabla\sigma, \mathrm{D}g, r} (\nabla\sigma)^\top \mathrm{diag}(v) W \mathrm{D}gr$$

$$\overset{(i)}{\leq} \max_{\|s\|_\infty \leq 1, \mathrm{D}g, r} s^\top \mathrm{diag}(v) W \mathrm{D}gr$$

$$\overset{(ii)}{\leq} 2 \max_{\|s\|_\infty \leq 1, \|t\|_\infty \leq 1} \epsilon s^\top \mathrm{diag}(v) W \mathrm{diag}(\lambda) t$$

In (i), we use the assumption that the non-linear activation function applied element-wise has bounded gradients $\nabla\sigma \in [0, 1]$, which holds in many settings (e.g., for ReLU, $\nabla\sigma \in [0, 1]$; for sigmoid, $\nabla\sigma \in [0, 1/4)$). In (ii), we use the following results. As $\|Mr\| = \epsilon$ defines an ellipsoid $\mathcal{E}$ centered at 0, $\mathrm{diag}(z)r$ with $\|z\|_\infty \leq 1$ corresponds to the union of the set of ellipsoids mirrored to $\mathcal{E}$ along the axises. We can thus bound $\mathrm{diag}(z)r$ using the axis aligned bounding box (AABB) of these ellipsoids. We have the following proposition (proof in appendix):

**Proposition 2.** *The AABB of the ellipsoid defined by $\|Mr\| = \epsilon$ is given by $[-\epsilon\lambda_i, \epsilon\lambda_i]$ $(1 \leq i \leq d)$, where $\lambda_i^2$ is the $i$-th diagonal element of the matrix $(M^\top M)^{-1}$.*

Because $\mathrm{diag}(v)W\,\mathrm{diag}(\lambda)$ is not necessarily positive (negative) semidefinite, the above formulation is in general a non-convex optimization problem, which is similar to the NP-hard MAXCUT problem. We thus resort to the Semidefinite Programming (SDP) relaxation.

We first re-parameterize the variables as:

$$z \stackrel{\mathrm{def}}{=} \begin{bmatrix} s \\ t \end{bmatrix}$$

$$P \stackrel{\mathrm{def}}{=} \begin{bmatrix} 0 & \mathrm{diag}(v)W\,\mathrm{diag}(\lambda) \\ \mathrm{diag}(\lambda)W^\top \mathrm{diag}(v) & 0 \end{bmatrix}$$

we obtain the formulation:

$$\ell_{\mathrm{adv}}(x,r) \leq \max_{||z||_\infty \leq 1} \epsilon z^\top P z$$

Using the fact that $z^\top P z = \mathrm{tr}(zz^\top P)$, we first have:

$$\max_{||z||_\infty \leq 1} z^\top P z = \max_{Z=zz^\top,\,||z||_\infty \leq 1} \mathrm{tr}(ZP)$$

By relaxing $Z = zz^\top$ and $\|z\|_\infty \leq 1$ with $Z \succeq 0$ and $\mathrm{diag}(Z) \leq 1$ (Boyd & Vandenberghe, 2004), we have the following convex SDP problem:

$$\max \mathrm{tr}(ZP)$$
$$\mathrm{s.t.} \begin{cases} \mathrm{diag}(Z) \leq 1 \\ Z \succeq 0 \end{cases}$$

which is efficiently solvable using off-the-shelf SDP optimizers.

### 4.2 EMPIRICAL EVALUATION

Next we empirically validate the above analytical results. Following the same setting as in §3, we mainly use MNIST (LeCun et al., 1998) as the benchmark dataset and a two layer neural network as the target network model $f$. We consider four variants of $f$: (i) $f_{\mathrm{nt}} - f$ is normally trained, (ii) $f_{\mathrm{at}} - f$ is adversarially trained (Madry et al., 2018), (iii) $f$ is trained with regularized bounds, and (iv) $f_{\mathrm{stn}} - f$ is trained with spatial transformer network. For (iii), we consider three types of bounds: Frobenius $- f_{\mathrm{fro}}$, spectral $- f_{\mathrm{spe}}$, and SDP $f_{\mathrm{sdp}}$ (our proposed method). The implementation details are presented in the appendix.

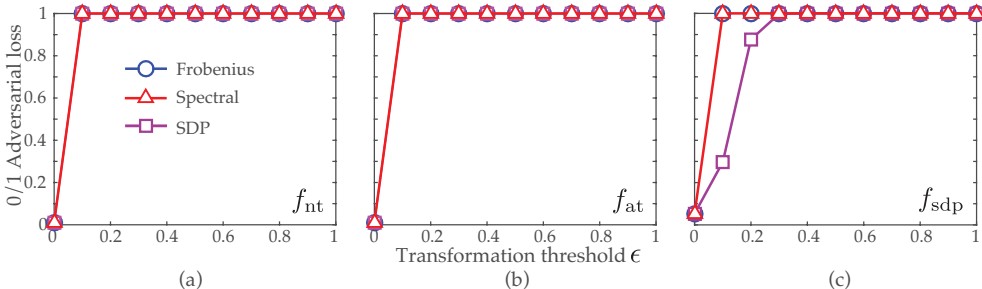

Figure 5: 0/1 adversarial loss for different variants of $f$ on MNIST based on Frobenius, spectral and SDP upper bound.

**Upper Bounds** We evaluate the different upper bounds on adversarial loss. For each of the networks described above, we compute the 0/1 loss based on the Frobenius, spectral, and SDP bounds respectively, with results shown in Figure 5. It is observed that all the bounds are fairly loose: the estimated adversarial loss goes to 1 as $\epsilon$ reaches 0.1. This again echos the impossibility results found in §3. However, our proposed SDP bound is slightly tighter than alternative bounds under $f_{\mathrm{sdp}}$.

**Model Robustness** We also observe that $f_{\mathrm{sdf}}$ is more robust than alternative models under varying perturbation budget. Specifically, we plot both the success rates of spatial transformation attacks and the test accuracy of different models under varying $\tau$ in Figure 6. Here $\tau$ controls the allowed magnitude of spatial transformation, with $\tau = 0.05$ used in (Xiao et al., 2018). For $f_{\mathrm{sdp}}$, it is observed that there are the reduction of around 0.1 for attack success rates and the gain of around 0.08 for adversarial test accuracy, compared with other training objectives.

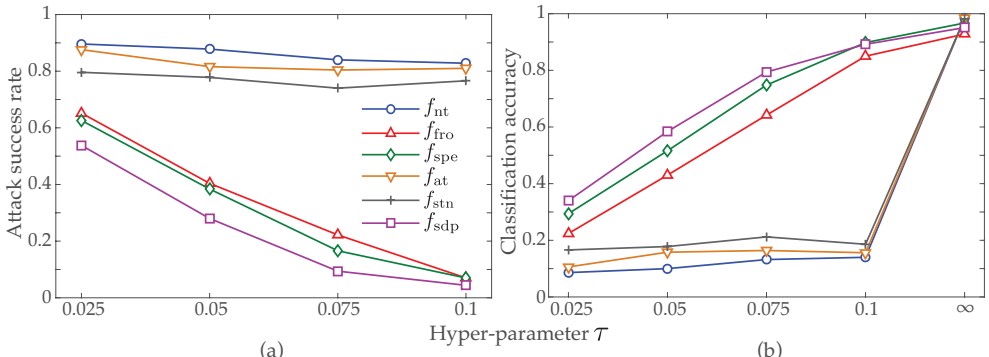

Figure 6: (a) Spatial transformation attack success rate versus $\tau$ under different training objectives. (b) Test accuracy of spatial transformed adversarial inputs for different training objectives and $\tau$ ($\tau = \infty$ corresponds to benign cases).

## 5 RELATED WORK

**Adversarial Inputs** Due to their increasing use in security-critical domains, machine learning models are becoming the targets of malicious attacks (Barreno et al., 2010; Biggio & Roli, 2018). Compared with simple models (e.g., decision tree, support vector machine, and logistic regression), securing deep neural networks deployed in adversarial settings poses even more challenges due to their significantly higher model complexity (LeCun et al., 2015). A variety of adversarial attacks have been proposed, including both white-box attacks (Szegedy et al., 2014; Goodfellow et al., 2015; Moosavi-Dezfooli et al., 2015; Papernot et al., 2016a; Kurakin et al., 2016; Carlini & Wagner, 2017; Meng & Chen, 2017; Xu et al., 2018) and black-box attacks (Papernot et al., 2016; Liu et al., 2016; Reddy Mopuri et al., 2017).

**Spatial Transformation** While most state-of-the-art attacks directly modify the pixel values of benign images, recent work (Xiao et al., 2018; Engstrom et al., 2017b) has proposed to use spatial deformation as an alternative to generate adversarial inputs. Compared with pixel perturbation, spatial transformation tends to better preserve the perceptual quality of original images. Yet, besides their perceptual superiority, thus far little is known about the security properties of spatially transformed adversarial inputs.

**Provable Defenses** Another line of research has focused on improving model resilience against adversarial attacks by developing new training and inference strategies (Goodfellow et al., 2015; Huang et al., 2015; Papernot et al., 2016b; Ji et al., 2018; Madry et al., 2018; Tramèr et al., 2018; Kannan et al., 2018). Yet, the fundamental challenges of defending against adversarial attacks stem from their adaptive nature. Existing defenses, once deployed, can be easily circumvented by adaptive attacks. This arms race between attacks and defenses has motivated the development of provable defenses (Bastani et al., 2016; Raghunathan et al., 2018; Wong & Zico Kolter, 2018), which ensure that for a given network, no attack is able to force the error to exceed a certain threshold. However, existing provable defenses all assume norm-bounded perturbation. It is an open question whether such defenses exist for spatial transformation-based attacks.

## 6 CONCLUSION

This work represents the first systematic study on provable defenses against spatial transformation-based adversarial attacks. Our findings include both possibility and impossibility results. We show that such provable defenses do not exist in practice. For any given networks, the adversary is able to find legitimate inputs and imperceptible transformations to generate adversarial inputs that force arbitrarily large errors. Yet, we also show that it is still possible to construct adversarial training methods to significantly improve the robustness of given networks against adversarial inputs over empirical datasets. We hope this work may inspire designing more effective defenses against spatial transformation-based attacks and adversarial attacks in general.

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

## APPENDIX

### 6.1 NOTATIONS AND SYMBOLS

| Symbol | Description |
|---:|:---|
| $x$ | genuine input |
| $\tilde{x}$ | adversarial input |
| $f(\cdot)$ | classifier |
| $r$ | flow vector |
| $g(r; x)$ | spatial transformation $r$ on $x$ |
| $\Delta f(x, r)$ | margin between $f \circ g(r; x)$ and $f(x)$ |
| $\nabla f(\cdot)$ | gradient of $f$ |
| $\mathrm{D}g(\cdot)$ | Jacobian matrix of $g$ |
| $\pi_2(\cdot)$ | $L_2$ projection $\pi_2(z) = \frac{z}{\|z\|}$ |
| $\pi_\infty(\cdot)$ | $L_\infty$ projection $\pi_\infty(z) = \frac{z}{\|z\|_\infty}$ |

Table 1: Symbols and notations.

### 6.2 PROOFS OF PROPOSITIONS, LEMMAS, AND THEOREMS

(Unless noted otherwise, we use $\| \cdot \|$ to indicate $L_2$ norm.)

PROOF OF PROPOSITION 3

**Proposition 3.** *The extreme values of Eqn (4) are achieved when $r^{(u)} = r^{(v)}$.*

*Proof.* (Proposition 3) Consider the first-order approximation of Eqn (4). Recall that $r = [r^{(u)}, r^{(v)}]$ and notice that $\nabla g(0) = [x, x]$ according to Eqn (2.2). We can rewrite the above expression as:

$$\ell_{\mathrm{adv}}(x, r) \quad = \quad \mathrm{tr}\left( (\mathrm{diag}(\nabla f(x)) [x, x]) \left[ r^{(u)}, r^{(v)} \right] \right)$$

$$\text{s.t.} \quad \begin{cases} \sqrt{r^{(u)\top} M^2 r^{(u)} + r^{(v)\top} M^2 r^{(v)}} \leq \epsilon \\ \|x\|_\infty \leq 1 \end{cases}$$

Due to the symmetry of $r^{(u)}$ and $r^{(v)}$, the extreme values of $\Delta f(x, r)$ is attained when $r^{(u)} = r^{(v)}$.

$\square$

*Proof.* (Proposition 1)

We derive the following matrix form for Eqn (2):

$$\ell_{\mathrm{diff}}(r) = \sqrt{2r^\top \left( \sum_{i \in \{t,b,l,r\}} \left( I - M^{(i)} \right)^\top \left( I - M^{(i)} \right) \right) r} \tag{9}$$

where $I$ denotes the identity matrix, and $\mathrm{tr}$ is the trace operator. We have the following proposition to simplify the notation (proof in Appendix).

We first prove that $\sum_{i \in \{l,r,t,b\}} (I - M^{(i)})^\top (I - M^{(i)})$ is diagonally dominant. It suffices to show that each matrix $(I - M^{(i)})^\top (I - M^{(i)})$ is diagonally dominant. Without loss of generality, assume $M^{(i)}$ corresponds to the top neighbor matrix.

Let $A = I - M^{(i)}$. Consider the $i$-th pixel. We differentiate three cases: (i) if $i$ is on the top boundary, it must be the top neighbor of another pixel $j$; (ii) if $i$ is on the bottom boundary, it must be the bottom neighbor of another pixel $k$; (iii) otherwise, $i$ is the top neighbor of some $j$ and the bottom neighbor of some $k$.

Consider the $i$-th column of $A$. We have: (i) $A_{ii} = 0$, $A_{ji} = -1$, and $A_{ri} = 0$ for $r \neq i, j$; (ii) $A_{ii} = 1$, $A_{ri} = 0$ for $r \neq i$; (iii) $A_{ii} = 1$, $A_{ji} = -1$, and $A_{ri} = 0$ for $r \neq i, j$. Now we compute $Z = A^\top A$. First consider the diagonal element $Z_{ii}$: (i) $Z_{ii} = 1$; (ii) $Z_{ii} = 1$; and (iii) $Z_{ii} = 2$. Then consider the off-diagonal element $Z_{ij}$: (i) if $i$ is $j$'s top neighbor, then $Z_{ij} = -1$, and $Z_{ij} = 0$ otherwise; (ii) if $i$ is $j$'s bottom neighbor, $Z_{ij} = -1$, and $Z_{ij} = 0$ otherwise; (iii) if $i$ is $j$'s top neighbor or bottom neighbor, $Z_{ij} = -1$, and $Z_{ij} = 0$ otherwise.

In all the cases, $Z_{ii} \geq 0$ and $|Z_{ii}| \geq \sum_{j \neq i} |Z_{ij}|$, i.e., $Z$ is diagonally dominant with real non-negative diagonal elements. Thus, $\sum_{i \in \{l,r,t,b\}} (I - M^{(i)})^\top (I - M^{(i)})$ is positive semidefinite. Using the spectral decomposition, we can rewrite it as: $\sum_{i \in \{l,r,t,b\}} (I - M^{(i)})^\top (I - M^{(i)}) = M^\top M$, where $M$ is a square matrix of the same rank as $\sum_{i \in \{l,r,t,b\}} (I - M^{(i)})^\top (I - M^{(i)})$.  □

PROOF OF THEOREM 2

*Proof.* Recall that $\|Mr\| = \epsilon$ defines an ellipsoid. It is straightforward to see that the supremum of $\tilde{\Delta} f(x, r) = v^\top \operatorname{diag}(x) r$ is attained only if $r$ lies on the ellipsoid boundary. Otherwise, suppose $r_0 = \arg\max_r \tilde{\Delta} f(x, r)$ and $\|Mr_0\| < \epsilon$. Note that $f(x, r) \geq 0$. We may extend $r_0$ to the ellipsoid boundary to further increase $\tilde{\Delta} f(x, r)$. Thus, we only need to consider the case of $\|Mr\| = \epsilon$. For convenience, we rewrite the ellipsoid in the following form: $r = M^{-1} s$ with $\|s\| = \epsilon$.

Thus, $\sup_{x,r} \tilde{\ell}_{\mathrm{adv}}(x, r) = \sup_{x,s} v^\top \operatorname{diag}(x) M^{-1} s$, which represents the maximum projection of $s$ on $M^{-1} \operatorname{diag}(x) v$. Note that $s$ lies on the ball of $\|s\| = \epsilon$ and that $M^{-1} \operatorname{diag}(x) v = M^{-1} \operatorname{diag}(v) x$. We have the formulation as follows:

$$\sup_{\|x\|_\infty \leq 1, \|Mr\| \leq \epsilon} \tilde{\ell}_{\mathrm{adv}}(x, r) = \epsilon \sup_{\|x\|_\infty \leq 1} \|M^{-1} \operatorname{diag}(v) x\| \tag{10}$$

which is attained by $s = \epsilon \pi_2(M^{-1} \operatorname{diag}(v) x)$.

It is noted that without the constraint of $\|x\|_\infty \leq 1$, the supremum above is attained by assigning $x$ the primary eigenvector $v^*$ of $\operatorname{diag}(v) M^{-2} \operatorname{diag}(v)$. We thus set $x = \pi_\infty(v^*)$, which leads to:

$$\sup_{\|x\|_\infty \leq 1, \|Mr\| \leq \epsilon} \tilde{\ell}_{\mathrm{adv}}(x, r) \geq \epsilon \|M^{-1} \operatorname{diag}(v) \pi_\infty(v^*)\|$$

□

PROOF OF THEOREM 1

*Proof.* Similar to the proof of Theorem 2, it is easy to see that the supremum of $\tilde{\Delta} f(x, r)$ is attained only if $r$ lies on the ellipsoid boundary of $\|Mr\| = \epsilon$. For convenience, we redefine the ellipsoid as: $r = M^{-1} s$ with $\|s\| = \epsilon$.

According to the definition, we have:

$$\sup_{\|x\|_\infty \leq 1, \|Mr\| \leq \epsilon} \tilde{\ell}_{\mathrm{adv}}(x, r) = \sup_{\|s\| \leq \epsilon, \|x\|_\infty \leq 1} (\nabla\sigma)^\top \operatorname{diag}(v) W \operatorname{diag}(x) M^{-1} s$$

As $s$ lies on the ball of $\|s\| = \epsilon$, for fixed $x$, we have:

$$\sup_{\|x\|_\infty \leq 1, \|Mr\| \leq \epsilon} \tilde{\ell}_{\mathrm{adv}}(x, r) = \epsilon \sup_{\|x\|_\infty \leq 1} \|M^{-1} \operatorname{diag}(x) W^\top \operatorname{diag}(v) \nabla\sigma\|$$

which is achieved by $s = \epsilon \pi_2(M^{-1} \operatorname{diag}(x) W \operatorname{diag}(v) \nabla\sigma)$.

We then define the following auxiliary function:

$$\phi(x) = \frac{1}{2} \|M^{-1} \operatorname{diag}(x) W^\top \operatorname{diag}(v) \nabla\sigma\|^2$$

Recall that in the case that $\sigma$ is ReLU, $(\nabla\sigma)_i = 1$ if $(Wx)_i > 0$ or $(\nabla\sigma)_i = 0$ otherwise. Observe that $\phi(x)$ has the absolute lower bound of 0, which can be achieved when $x = 0$, i.e., all the neurons in the first layer are inactive. We attempt to increase $\phi(x)$ by maximizing the number of active neurons.

For a practical neural network $f$, we may consider each column of $W$ as a basis function. We thus assume $W$ has full column rank, i.e., $\text{rank}(W) = d$. Therefore, by properly setting $x$, we may make at least $d$ dimensions of $Wx$ larger than 0. Let $W_d$ be the submatrix of $W$ that consists of $d$ independent rows of $W$. We consider $x_* = \pi_\infty(W_d^{-1}\mathbf{1})$ where $\mathbf{1}$ is an all-one vector. It is easy to verify that (i) $W_d x_* > 0$, i.e., $\nabla\sigma(W_d x_*) = 1$ and $\|x_0\|_\infty \leq 1$. That is, $x_*$ is a valid input and activates at least $d$ neurons in the first layer of the network. We have the following lower bound:

$$\sup_{\|x\|_\infty \leq 1, \|Mr\| \leq \epsilon} \tilde{\ell}_{\text{adv}}(x, r) \geq \epsilon \|M^{-1}\,\text{diag}(x_*)W^\top \text{diag}(v)\nabla\sigma(Wx_*)\|.$$

$\square$

## 6.3 Results for Linear Classifiers

## 6.4 Impossibility Results

In the case of single-layer neural network (i.e., linear classifier), we have $f(x) = v^\top x$, where $v$ is the weights of the network. In this case, we have the following result:

$$\ell_{\text{adv}}(x, r) = \int_0^1 v^\top \text{D}g(zr; x)r dz \tag{11}$$

To find $x_*$ and $r_*$, we consider a linear approximation of $\ell_{\text{adv}}(r; x)$:

$$\tilde{\ell}_{\text{adv}}(x, r) = v^\top \text{D}g(0; x)r = v^\top \text{diag}(x)r$$

where $\text{D}g(0; x) = \text{diag}(x)$ according to Eqn (1). Theorem 2 finds a lower bound of the supremum of $\tilde{\ell}_{\text{adv}}(x, r)$ (proof in appendix).

**Theorem 2.** *For a single-layer network $f$, we have*

$$\sup_{\|x\|_\infty \leq 1, \|Mr\| \leq \epsilon} \tilde{\ell}_{\text{adv}}(x, r) \geq \epsilon \|M^{-1}\,\text{diag}(v)x_*\|$$

*where $x_* = \pi_\infty(v^*)$ and $v^*$ represents the primary eigenvector of $\text{diag}(v)M^{-2}\,\text{diag}(v)$. The bound is attained when*

$$r_* = \epsilon M^{-1}\pi_2(M^{-1}\,\text{diag}(v)x_*). \tag{12}$$

While Theorem 2 finds high-quality setting of $(x_*, r_*)$, we may further increase the adversarial loss by approximating Eqn (11). Let $r^{(i)} = \frac{2i-1}{2n}r$. We compute the midpoint approximation of Eqn (11):

$$\ell_{\text{adv}}(x, r) \approx \frac{1}{n}\underbrace{\sum_{i=1}^n v^\top \text{D}g\left(r^{(i)}; x\right) r}_{\text{Summation (i)}} \tag{13}$$

where we divide the interval $[0, 1]$ into $n$ subintervals.

**Updating $x_*$**

We first update $x_*$ to maximize Eqn (13) With fixed $r$. To this end, we consider $x$ as a symbolic vector in Eqn (13). Let $\alpha_i$ be the coefficient of $x$'s $i$-th element in Eqn (13). We set $x_* = [\text{sign}(\alpha_1), \text{sign}(\alpha_2), \ldots, \text{sign}(\alpha_d)]^\top$, which maximizes Eqn (13) under fixed $r$ and the constraint of $\|x\|_\infty \leq 1$.

**Updating $r_*$**

Let $r_k$ be the setting of $r_*$ at the $k$-th iteration. With $x_*$ fixed, we may update $r_*$ as follows:

$$r_{k+1} = \epsilon M^{-1}\pi_2\left(M^{-1}\sum_{i=1}^n \text{D}g\left(r_k^{(i)}; x_*\right) v\right) \tag{14}$$

Intuitively, to maximize Eqn (13), we find $r_{k+1}$ that aligns with the direction of the summation (i) in Eqn (13) under $\|Mr_{k+1}\| \leq \epsilon$.

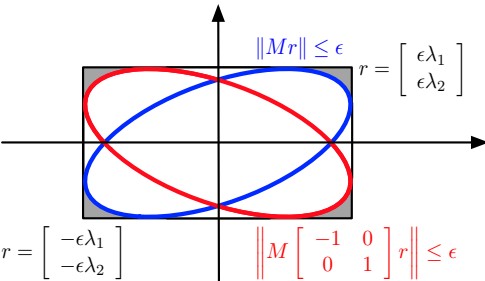

Figure 7: Bounds of $\mathrm{diag}(s)r$, where the dashed area corresponds to $\mathrm{diag}(s)r$ that satisfies $||Mr|| \leq \epsilon$ and $||s||_\infty \leq 1$.

**Attack**

Algorithm 2 sketches the attack against a given single-layer network. After initialization (line 1-3), it updates $x_*$ and $r_*$ in an interleaving manner (line 4-7).

---
**Algorithm 2:** Attack against single-layer networks.

---
**Input**: $v$: $f$'s weights; $\epsilon$: threshold of perturbation magnitude; $M$: neighboring matrix; $n$: parameter of midpoint approximation
**Output**: $x_*$: genuine input; $r_*$: spatial transformation
```
// initialization
```
1   $v^* \leftarrow$ primary eigenvector of $\mathrm{diag}(v)M^{-2}\,\mathrm{diag}(v)$;
2   $x_* \leftarrow \pi_\infty(v^*)$;
3   set $r_*$ according to Eqn (12);
4   **while** *not converged* **do**
```
      // refinement of x*
```
5      compute Eqn (13) with symbolic $x$;
```
      // α_i:  coefficient of x's i-th dimension in Eqn(13)
```
6      $x_* \leftarrow [\mathrm{sign}(\alpha_1), \mathrm{sign}(\alpha_2), \ldots, \mathrm{sign}(\alpha_d)]^\top$;
```
      // refinement of r*
```
7      update $r_*$ according to Eqn (14);
8   return $(x_*, r_*)$;

---

### 6.5 POSSIBILITY RESULTS

In the case of single-layer network (i.e., linear classifier), we have:

$$
\begin{aligned}
\ell_{\mathrm{adv}}(x,r) &\leq \max_{z\in[0,1],\|Mr\|\leq\epsilon} v^\top \mathrm{D}g(zr;x)r \\
&\leq 2 \max_{\|s\|_\infty\leq 1,\|Mr\|\leq\epsilon} v^\top \mathrm{diag}(s)r
\end{aligned}
\tag{15}
$$

where the last inequality follows from the fact that as defined in Eqn (1), under bilinear interpolation, the diagonal elements of $\mathrm{D}g(zr;x)$ vary within the interval of $[-2,2]$.

Let $\lambda = [\lambda_1, \lambda_2, \ldots, \lambda_d]^\top$. We have the following derivation of Eqn (15) (note that we re-parameterize $r$ with $t$):

$$
\begin{aligned}
\ell_{\mathrm{adv}}(x,r) &\leq 2 \max_{\|t\|_\infty\leq 1} \epsilon v^\top \mathrm{diag}(\lambda)t \\
&= 2\epsilon \left\|v^\top \mathrm{diag}(\lambda)\right\|_1
\end{aligned}
\tag{16}
$$

where the upper-bound is attained when $t = \mathrm{sign}(v^\top \mathrm{diag}(\lambda))$.

Note that this upper bound is irrelevant to the original input $x$ and solely dependent on the network's properties (i.e., $v$ and $\lambda$). Thus, it only needs to be computed once for a given network.

## 6.6 EXTENSION TO MULTI-CLASS CASES (IMPOSSIBILITY)

We can extend Algorithm 2 and Algorithm 1 for a $K$-class classifier $f(\cdot)$. We considerthe CW attack and take $\kappa = 0$:

$$\ell_{\mathrm{adv}}(r) = \max_{y \neq \tilde{y}} \sigma_y(g(r;x)) - \sigma_{\tilde{y}}(g(r;x)) \qquad (17)$$

For the most popular gradient-based adversarial attack methods, their update rules of $r$ at $t$-th iteration are:

$$r_{t+1} = r_t - \alpha \frac{\partial \ell_{\mathrm{adv}}}{\partial r}(r) \qquad (18)$$

where $\alpha$ is learning rate. Let $y^\star$ such that $\arg \max_{y \neq \tilde{y}} \sigma_y(g(r;x)) = y^\star$, then according to the chain rule, we have $\frac{\partial \max_{y \neq \tilde{y}} \sigma_y \circ g}{\partial r}(r;x) = \frac{\partial \sigma_{y^\star} \circ g}{\partial r}(r;x)$. Then we derive a simple work around: at each iteration, we only update based on the two classes case between class $y^\star(-)$ and $\tilde{y}(+)$.

## 6.7 EXTENSION TO MULTI-CLASS CASES (POSSIBILITY)

We extend the proposed SDP training objective to multiple classes case. Here we only consider a two-layer neural network, it is straightforward to apply the extension to a one-layer network. Similar to Eqn (9), we define:

$$P^{ij} \overset{\mathrm{def}}{=} \begin{bmatrix} 0 & \mathrm{diag}(v_i - v_j)W\,\mathrm{diag}(\lambda) \\ \mathrm{diag}(\lambda)W^\top \mathrm{diag}(v_i - v_j) & 0 \end{bmatrix}$$

for all pair of classes $i(+), j(-)$. Then we have

$$\ell_{\mathrm{adv}}^{ij}(x,r) \leq \max_{\mathrm{diag}(Z) \leq 1, Z \succeq 0} \mathrm{tr}(ZP^{ij}) \overset{\mathrm{def}}{=} q^{ij}(x)$$

For an image $x$ with class $y$, the attack is failed if and only if $\max_{i \neq y} \ell_{\mathrm{adv}}^{iy}(x) \leq 0$. Therefore, we take $\max_{i \neq y} q^{iy}(x)$ as the upper bound.

## 6.8 ADDITIONAL EXPERIMENTAL RESULTS

### IMPLEMENTATION DETAILS

For the empirical evaluation (ii), we follow the experimental setup in Raghunathan et al. (2018) and evaluate the part with a two-layer network $f$ with 500 hidden units. We construct a set of variants of $f$ using different training methods. We report our training objectives and details below:

1. $f_{\mathrm{nt}} - f$ is normally trained, with cross-entropy loss and without explicit regularization.

2. $f_{\mathrm{fro}} - f$ is trained with Hinge loss Frobenius norm regularization (Fro-NN) and $\lambda (\|W\|_F + \|v\|_2)$ with $\lambda = 0.08$.

3. $f_{\mathrm{spe}} - f$ is trained with hinge loss and a regularizer $\lambda (\|W\|_2 + \|v\|_2)$ with $\lambda = 0.09$.

4. $f_{\mathrm{stn}} - f$ is trained with hinge loss and a three layer spatial transformer network with 100 hidden units processes input images before feeding them to classification network.

5. $f_{\mathrm{at}} - f$ is trained with cross-entropy loss with an adversarial training regularization against spatial transformed adversarial examples. We take regularization coefficient $\lambda_r = 0.3$ for training and $\lambda = 0.05$ for regularizing the $\ell_{\mathrm{diff}}$. Due to Xiao et al. (2018) taking L-BFGS for solving the optimization problem, it is very slow to generate adversarial examples on the fly. Instead, we generate a pool of adversarial examples every $k = 3$ epochs, and sample adversarial example from pools during training.

6. $f_{\mathrm{sdp}} - f$ is the upper bound we derived in Eqn (9), and we follow the implementation of SDP-NN in Raghunathan et al. (2018). In particular, we also optimize the dual form of Eqn (9) so that we can backpropagate through both the classification loss and the SDP regularization term.

The other naive bounds used in the § 4:

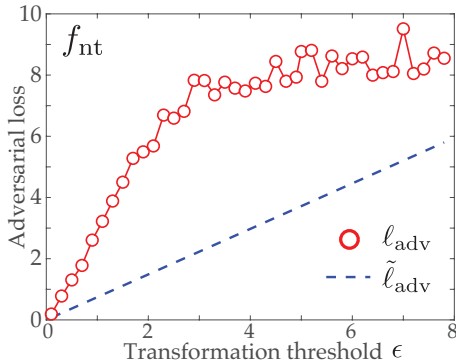

Figure 8: Lower bounds estimated with respect to different networks.

1. Spectral bound

$$\ell_{\text{diff}}(x, r) \leq \epsilon \|W\|_2 \|v\|_2$$

2. Frobenius bound

$$\ell_{\text{diff}}(x, r) \leq \epsilon \|W\|_F \|v\|_2$$

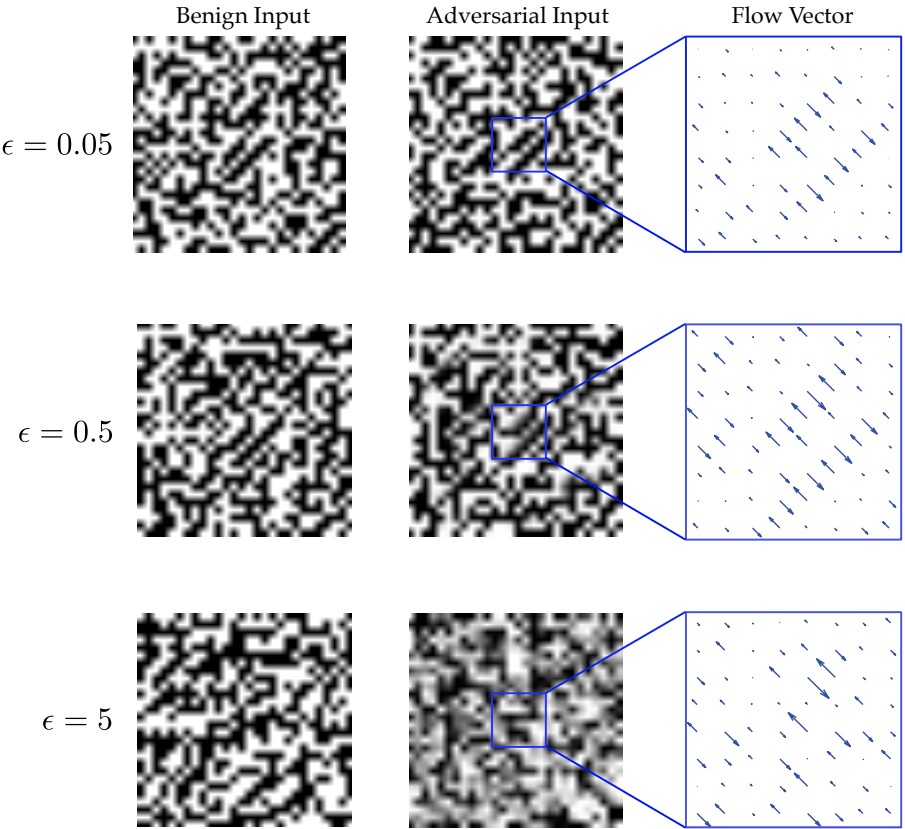

Figure 9: Samples of benign and adversarial inputs found by Algorithm 1 under $f_{\text{rd}}$.

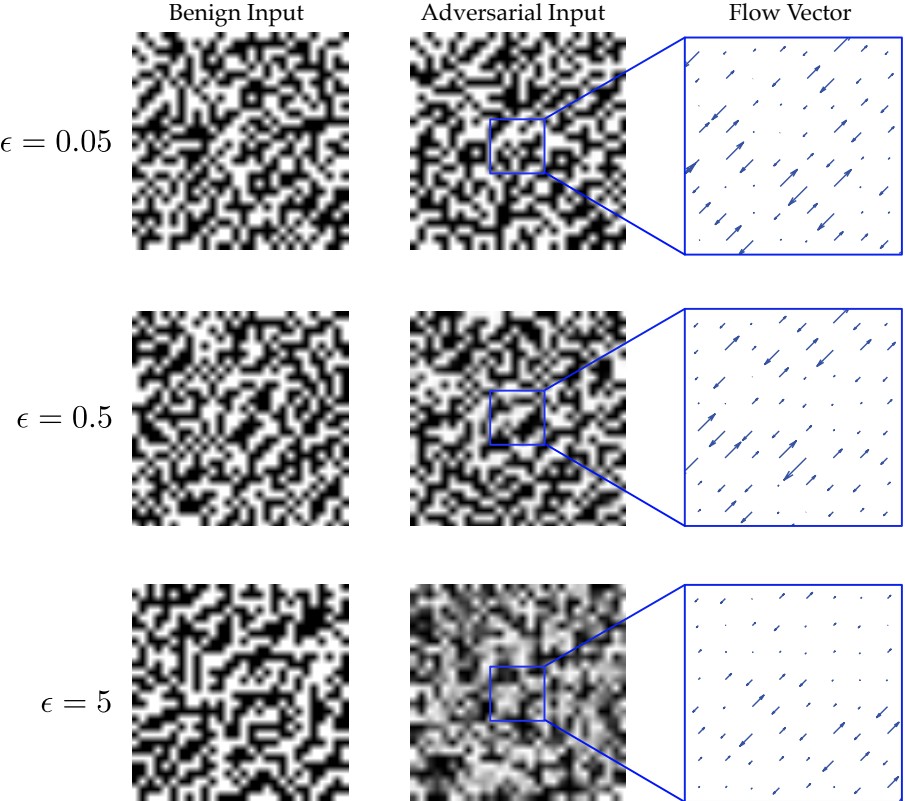

Figure 10: Samples of benign and adversarial inputs found by Algorithm 1 under $f_{\mathrm{at}}$.

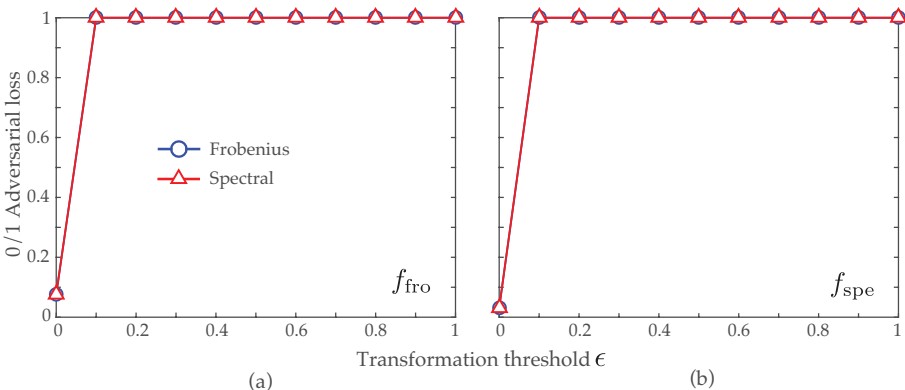

Figure 11: Upper bounds on 0/1 adversarial loss for different networks on MNIST.

