# OpenReview forum: "Provable Defenses against Spatially Transformed Adversarial Inputs: Impossibility and Possibility Results"
_ICLR.cc/2019/Conference_

### Official Review · AnonReviewer2 · 2018-11-01
**Can the proposed defense is secure against pixel-based AEs?**

**Rating:** 5
**Confidence:** 3

**Review:**

The presented analysis well characterizes the behavior of the spatially transformed adversarial inputs and the proposed defense is empirically confirmed to achieve more accurate and robust classification under attacks.

One concern is that the defender cannot learn whether the adversary employs spatially transformed AEs or pixel-based AEs (or some others). What happens if the classifier trained with the proposed defense accept pixel-based AEs? I recommend the authors to associate spatially transformed AEs with pixel-based AEs to learn whether the proposed defense performs more robustly compared to existing defenses. If the proposed defense method performs well for spatially transformed AEs but is vulnerable to pixel-based AEs, it is useless.

It should be better to discuss more on computational efficiency of the proposed defense since it contains SDP solving. Is the proposed deense works with larger datasets such as CIFAR100 or ImageNet?

---

### Official Review · AnonReviewer1 · 2018-11-02
**Limited novelty compared to earlier work; poor presentation of results and conceptual differences between the proposed spatial transformation attack model vs. existing lp norm attack model**

**Rating:** 3
**Confidence:** 4

**Review:**

Summary: The paper studies a new attack model based on spatial transformations. The authors first formalize an attack model based on spatial transformation and then study attacks and defenses for this model.

Clarity: While the paper studies an important problem -- it's important to move out of the norm ball based attack models and consider different attacks like spatial transformations, in the current version, the presentation lacks clarity in both the formulation of the attack model, attacks, defenses and explanation of the results. For example, the impossibility result isn't clear: the claim is that any classifier has adversarial spatial transformations that are successful in causing misclassifcation for some threshold on the size of transformation. There is no explanation of how large this threshold is in practice. Is it small enough to be called an "impossibility result"? What does this threshold intuitively depend on?

Originality: The key contribution seems to be the formalization of some notion of spatial transformation. However, the final expression (Proposition 1) basically looks just like an l_p norm but after transforming it by some "fixed" matrix M. The expressions for this new attack model where || M r|| < \eps for some perturbation \eps look pretty similar to the case previously considered (where M was essentially identity). For example, Raghunathan et al. 2018 and Hein & Andriushchenko 2017. The paper is also missing discussion on the structure of this matrix M, and how it changes the attacks and defenses in practice

Significance: I think the problem of spatial transformation based adversarial examples is important and the authors have the right goals. However, the current presentation makes it hard to understand the main results provided and hence I would rate that the contribution is not very significant.

Overall: I highly recommend the authors to revise the presentation and clarify a) the main conceptual differences of the new attack model (matrix M of proposition 1) b) Formalize the impossibility and possibility results carefully with concrete theoretical/empirical results to back the claims

---

### Official Review · AnonReviewer3 · 2018-11-04
**Review comments**

**Rating:** 5
**Confidence:** 3

**Review:**

This paper proposed a defense against spatially transformed adversarial inputs and give the two main results on possibility (still possible to construct adversarial training methods to improve robustness) and impossibility (always exist spatially-transformed adversarial examples for any given networks and thus no certified defense)

The topic of studying certified defenses on adversarial examples is important, and I think the direction of dealing with spatially-transformed adversarial examples is interesting. However, this paper only analyze a simple one hidden layer neural network and the technique (e.g. sec 4, possibility result) does not seem to easily scale to deeper networks and networks with other types of layers (e.g pooling layers). Also,

I also feel the clarity of the paper should be improved.

Here are some questions:
1. Are there other metrics to measure spatial transformation? For the current setting as introduced in sec 2.1, it looks like there is no a uniform spatial transformation on the full image but rather different transformation applied on different local areas. Does it make more sense to say rotate the full image by some angle or shift it by some distance?

2. What is the pi_infty and pi_2 in Theorem 1? Why is it called Lower bound attack in sec 3.1?

3. What is the difference between f_fro, f_spe and f_sdp?

4. In Figure 6 (b), is the classification accuracy the nominal test accuracy of a classifier? If so, then the accuracy is too low (<90% for mnist) and thus considering the corresponding attack rates (Fig 6(a)) on these models are not meaningful. Please explain.

---

### Meta-Review · Area_Chair1 · 2018-12-16
**The clarity of presentation and evaluation is a concern**

**Confidence:** 5
**Recommendation:** Reject

**Metareview:**

This paper conducts a study on provable defenses to spatially transformed adversarial examples. In general, the paper pursues an interesting direction, but reviewers had many concerns regarding the clarity of the presentation and the depth of the experimental results, which the authors did not address in a rebuttal.